# Symmetric Multi-level Gradient-Inverse Consistency Network for Brain Image Registration with Large Deformation

**Haoying, Bai**[1]                                                                    HAOYINGBAI@MAIL.BNU.EDU.CN
**Tongtong, Che**[*1]                                                                          CHE@BNU.EDU.CN
**Jichang, Zhang**[1]                                                          ZHANGJICHANG@MAIL.BNU.EDU.CN
**Shuyu, Li**[*1]                                                                           SHUYULI@BNU.EDU.CN
[1] *State Key Laboratory of Cognitive Neuroscience and Learning*
*Beijing Normal University, Beijing, 100875, China.*

**Editors:** Accepted for publication at MIDL 2025

## Abstract

Accurate and robust deformable image registration is crucial for brain image analysis. While deep learning has significantly advanced this field, existing methods often lack robustness for large deformations due to inter-subject variability, frequently requiring pre-registration and relying heavily on data-driven approaches. To address these limitations, we propose an end-to-end **Symmetric Multis-level Gradient-Inverse Consistency Network (SM-GICNet)** for accurate and robust brain image registration. SM-GICNet employs 1) a symmetric multi-level framework with an attention gate mechanism to capture complex deformations at multiple scales, 2) a symmetric registration strategy at each level to mitigate directional bias, and 3) a gradient inverse consistency strategy to reduce reliance on data-driven constraints and control deformation field complexity. Experimental results demonstrate that our method is able to eliminate the need for pre-registration and outperforms state-of-the-art methods on large deformation registration tasks, on two datasets achieving a Dice similarity coefficient of 0.797 and 0.794. The implementation of our SM-GICNet is available online at https://github.com/LSYLAB/SM-GICNet.git.

**Keywords:** Symmetric registration, Consistency-Constrained, Inverse-Consistent, Multi-level

## 1. Introduction

Deformable image registration is a fundamental task in medical image analysis, aiming to establish a nonlinear spatial correspondence between a pair of images (source/moving and target/fixed images)(Sotiras et al., 2013). Large deformation image registration refers to the scenario where significant shape and positional differences exist between images, particularly in cases of high inter-subject heterogeneity or notable pathological variations(Meng et al., 2024; Wang et al., 2024). Achieving precise registration typically requires complex nonlinear transformations. Specifically, the registration process generally begins with an affine transformation for coarse alignment to capture large-scale deformations, followed by nonlinear transformations for fine alignment to optimize local details(Mok and Chung, 2022a).

---

* Corresponding authors

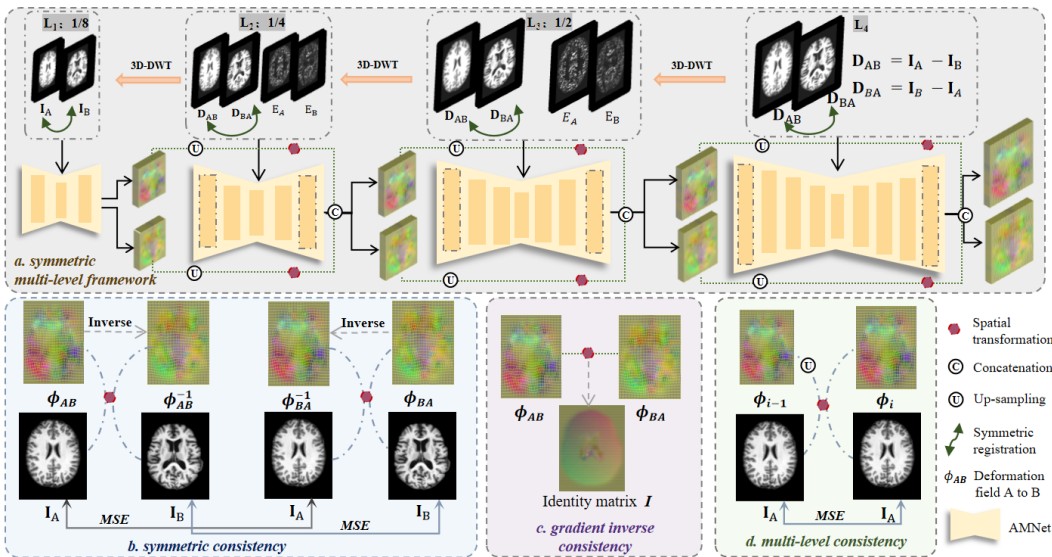

Figure 1: (a) A symmetric multi-level registration framework incorporating an attention gate mechanism; (b) symmetric deformation field consistency strategy at each level; (c) a deformation field constraint based on gradient inverse consistency; and (d) a multi-level consistency strategy. $D_{AB}$ means the intensity difference map between $I_A$ and $I_B$, $E_A$ and $E_B$ mean the energy image obtained from 3D-DWT. The (b)(c)(d) component serves as a loss constraint for our primary network (a), applied during the deformation field generation process at each layer.

Traditional registration methods typically rely on iterative optimization strategies to maximize similarity metrics in the transformation space (Oliveira and Tavares, 2014; Shen and Davatzikos, 2002). While these methods achieve relatively stable performance, they suffer from low computational efficiency and limitations when handling complex deformation fields. Recently, deep learning-registration methods achieve significant performance improvements by leveraging large datasets and the powerful modeling capabilities of neural networks (Wang et al., 2023, 2022; Shi et al., 2022). These methods directly predict the deformation field via neural networks, demonstrating superior performance in nonlinear registration tasks. However, they often focus solely on nonlinear registration, and their performance remains unsatisfactory without rigid or affine pre-registration (Mok and Chung, 2022b; Balakrishnan et al., 2019).

For nonlinear transformations, current deep-learning-based registration methods still face the challenges. In common multi-level image registration approaches, this is typically addressed with a coarse-to-fine registration strategy (Hering et al., 2019; Mok and Chung, 2020; Eppenhof et al., 2019). Specifically registration proceeds from a coarse alignment that captures large deformations to a fine alignment that incorporates local details. Unfortunately, the fixed directionality in existing multi-level methods inevitably leads to asymmetric and biased mapping, which may negatively impact subsequent image analysis tasks such

as computational anatomy. Some symmetric single-level registration methods have been proposed to simultaneously predict forward ($I_A \rightarrow I_B$) and backward ($I_B \rightarrow I_A$) transformations between images (Xu et al., 2023; Chen et al., 2023; Tian et al., 2024), enhancing the invertibility and accuracy of the deformation field. However, these methods mainly rely on similarity-driven optimizations, leading to slow convergence and high computational costs.

To address these issues, we propose a **Symmetric Multis-level Gradient-Inverse Consistency Network (SM-GICNet)** capable of directly handling large deformation registration tasks with high variability without pre-registration. Specifically, we introduce a method that combines a **symmetric multi-level** registration framework with an **attention gate mechanism** to capture deformation features at different scales and focus attention on complex deformation regions during the high-resolution stage. At each level, we employ a **symmetric deformation field consistency strategy**, where images A and B are simultaneously registered as each other's moving image, predicting both forward and backward deformation fields to ensure the stability and consistency of the transformations. Furthermore, we incorporate a **gradient inverse consistency constraint** to directly regularize the gradient alignment of the forward and backward deformation fields, limiting the complexity of the deformation field and mitigating reliance on purely data-driven optimizations.

Our main contributions are as follows:

- Our SM-GICNet is a symmetric multi-level registration framework with attention-gate mechanism for learning a progressively refined representations transformations, which eliminates the bias of generic directional image registration.

- Our SM-GICNet embraces the gradient inverse consistency constraint to replace conventional regularizers, which reduces reliance on purely data-driven optimization.

- Our SM-GICNet achieves accurate and fast registration without any pre-registration (rigid or affine registration), which demonstrates superior performance in large deformation registration tasks with high heterogeneity in brain MRI.

## 2. Related Work

### 2.1. Symmetric Diffeomorphic Registration

Symmetric registration is crucial for accurate medical image registration, particularly in estimating deformations between image pairs, improving geometric consistency and precision (Greer et al., 2021). Early methods independently estimated forward and backward transformations, lacking guaranteed inverse consistency (Zheng et al., 2021; Kim et al., 2021). Most deformable methods use displacement fields (Brauwers and Frasincar, 2021; Cao et al., 2018; Yang et al., 2017), neglecting differential properties like topology preservation and invertibility (Tian et al., 2025), hindering true symmetry. Diffeomorphic registration, using stationary velocity fields, offers a solution (Avants et al., 2008), ensuring smooth, invertible mappings. The diffeomorphic deformation field, $\phi_t$ (parameterized by $t \in [0,1]$), is generated from the velocity field as:

$$\frac{d\phi_v}{dt} = v_t(\phi_t) = v_t \circ \phi_t \tag{1}$$

Diffeomorphic models are advantageous for constructing symmetric registration networks due to their inherent invertibility.

## 2.2. Attention Gate Mechanism

The Attention Gate Mechanism (AGM) is an attention mechanism readily integrated into various Convolutional Neural Networks (CNNs) to enhance performance in tasks such as image segmentation, object detection, and image classification (Azad et al., 2024; Guo et al., 2022; Brauwers and Frasincar, 2021). By focusing on salient regions, AGMs guide networks to prioritize significant features (Li et al., 2023; Ranjbarzadeh et al., 2021). Numerous studies have shown that attention gate mechanisms significantly improve the accuracy of registration models in local regions, particularly in tasks requiring attention to fine anatomical structures (Chen et al., 2022; Tang et al., 2022). Attention gate networks based on U-Net architectures are especially prevalent in image registration. For example, Attention U-Net (Oktay et al., 2018) incorporates attention gate modules to focus on key regions of the input image, enhancing the model's response to specific areas while reducing interference from background noise and irrelevant features. This approach has demonstrates performance improvements on several public medical image datasets.

## 3. Methods

We present a symmetric multi-level gradient inverse consistency network (SM-GICNet) for large deformation image registration. As illustrated in Figure 1, SM-GICNet includes: (1) a symmetric multi-level registration framework incorporating the attention gate mechanism; (2) symmetric deformation field consistency strategy at each level; and (3) a deformation field constraint based on gradient inverse consistency.

### 3.1. Symmetric Multi-level Registration Framework

A novel symmetric multi-level registration framework with attention gate mechanism is proposed to effectively capture multi-scale deformation fields between image pairs. The multi-level architecture comprises four consecutive levels, efficiently capturing both global and complex local deformations within a single forward process. Symmetry is achieved by alternately using each image as the moving image in a single registration, yielding $\phi_{AB}$ and $\phi_{BA}$. Simultaneously, both forward ($\phi_{AB}$) and inverse ($\phi_{BA}^{-1}$) deformation fields are directly obtained for each image, enabling multi-constraint network learning.

The multi-level network is constructed using a 3D discrete wavelet transform (3D-DWT) (Ghasemzadeh and Demirel, 2018) to leverage both low-frequency global and high-frequency local information, following the input method of AMNet (Che et al., 2023). An AGM is introduced at level 4 to automatically enhance deformation field learning in crucial regions while suppressing background influence, improving the model's ability to deal with large deformations and get finer structural details. Further details regarding the attention gate network architecture are provided in Appendix A.

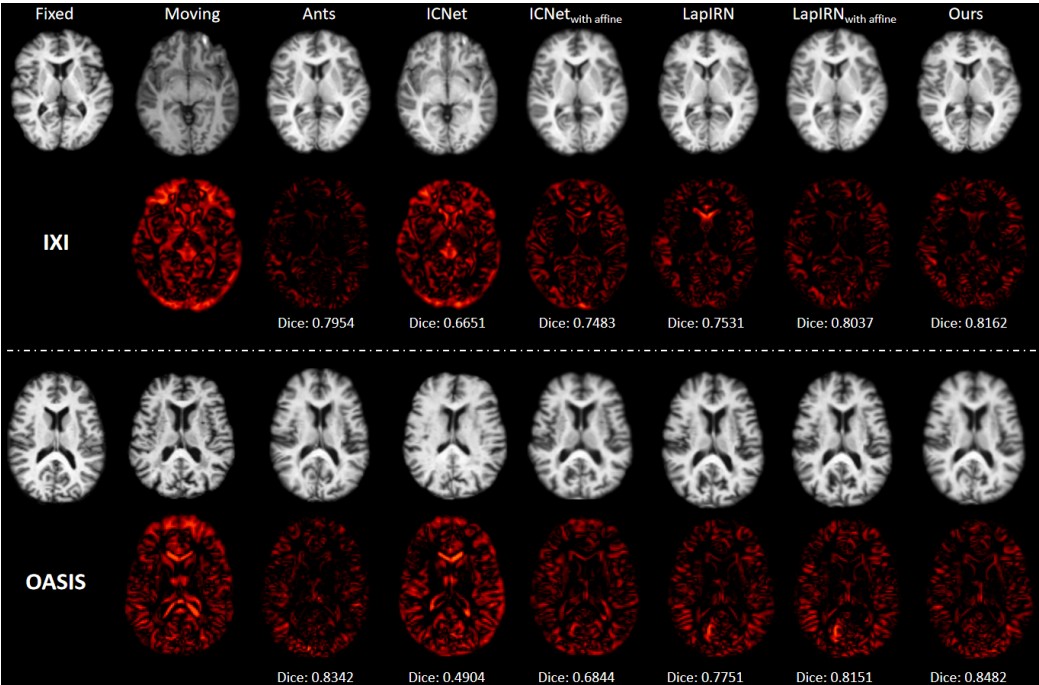

Figure 2: Qualitative registration results by three different methods and SM-GICNet on two datasets. The second and forth rows show the absolute intensity difference between the registered images and the fixed image, highlighting the residual misalignment after registration. The values down of warped images are the Dice score (DSC) between the warped image and the fixed image.

### 3.2. Symmetric Deformation Field Consistency Strategy

Our network employs a symmetric deformation field consistency strategy at each level, promoting bidirectional symmetry during unidirectional registration to enhance deformation field stability and consistency. Following diffeomorphic principles, a stationary velocity field is used instead of a displacement field for parameterization. The deformation field is defined as in the equation1. The velocity field $v$ is integrated over a unit time using a scaling and squaring operation with a time step $T = 7$ to obtain the final deformation field $\phi(1)$. The diffeomorphic model adapts to large or complex deformations, and since the output is a velocity field, the inverse velocity field, and subsequently the inverse deformation field, can be obtained by negating the velocity field. This forms the basis of the symmetric registration network.

Specifically, assuming the network learns the deformation field $\phi_{A \to B}$ from moving image $A$ to fixed image $B$, the inverse deformation field $\phi_{A \to B}$ is derived using diffeomorphic properties. The network then learns $\phi_{B \to A}$ from $B$ to $A$, ensuring consistency between these two deformation fields. This improves the inverse representation capabilities during forward registration.

### 3.3. Gradient Inverse Consistency Constraint

In a single registration, images $I_A$ and $I_B$ are registered reciprocally, alternately serving as the moving image. This yields both forward ($\phi_{AB}$) and backward ($\phi_{BA}$) deformation fields. To reduce data-driven dependence, the gradient inverse consistency constraint(Tian et al., 2023) is applied directly to the deformation fields. Unlike traditional registration methods that are heavily dependent on image similarity, this approach directly enforces constraints on the deformation fields based on their mathematical properties. This ensures that the matrix resulting from the forward deformation field and its inverse is consistent with the identity grid $\mathbf{I}$, suppressing unreasonable complexities in the deformation field, which not only improves robustness to image noise, contrast variations, and modality differences, but also accelerates convergence by reducing computational overhead during optimization.

### 3.4. Losses and Objectives

Our method employs a multi-component loss function to optimize image registration. The total loss is a weighted sum of the following terms.

**Consistency Loss**: This term enforces consistency between forward and inverse transformations. It comprises three sub-components:

**Symmetric Consistency**:

$$\mathcal{L}_{Sy} = \mathcal{L}_{MSE}(I_A \circ \phi_{AB}, I_B \circ \phi_{BA}) \tag{2}$$

**Inverse Consistency**:

$$\mathcal{L}_{In} = \mathcal{L}_{MSE}(\phi_{AB} \circ \phi_{AB}^{-1}), \mathbf{I}) \tag{3}$$

**Multi-level Consistency**:

$$\mathcal{L}_{Mu} = \mathcal{L}_{MSE}(\mathbf{w}_i, (\mathbf{w}_i + \mathbf{w}_{i-1})/2) \tag{4}$$

And the Consistency Loss is the sum of them: $\mathcal{L}_{consistency} = \lambda_s \mathcal{L}_{Sy} + \lambda_i \mathcal{L}_{In} + \lambda_m \mathcal{L}_{Mu}$, where $\lambda_s = 0.001$, $\lambda_i = 0.01$ and $\lambda_m = 0.0005$.

**Multi-level NCC Similarity Loss**: This term maximizes the Normalized Cross-Correlation (NCC) similarity between warped and fixed images across multiple resolution levels. Let $L_{NCC}^i$ denote the NCC loss at level i. Then: $L_{NCC} = \sum_i \alpha_i L_{NCC}^i$, where $\alpha_i$ are weights assigned to each level.

**Smoothness Regularization Loss**: This term penalizes overly complex deformations by minimizing the L2 norm of the deformation field gradients: $L_{smooth} = \sum_i \beta_i L_2^i$, where $L_2^i$ is the loss at level i and $\beta_i$ are weights.

The total loss function is given by: $L_{total} = L_{consistency} + L_{NCC} + L_{smooth}$.

## 4. Experiment

### 4.1. Data

We evaluate our method using the IXI[1] and OASIS[2] brain MRI datasets, which comprise 3D MRI scans weighted T1. From the IXI dataset, a subset of 314 subjects is divided into

---

1. https://brain-development.org/ixi-dataset/
2. https://sites.wustl.edu/oasisbrains/

training (n = 269), validation (n = 15) and testing (n = 30) sets. For the OASIS dataset, 360 subjects are divided into training (n=280), validation (n=20), and testing (n=60) sets. Our experiments focus on inter-subject registration, where moving and fixed image pairs are derived from both datasets. For the IXI test set (30 images), 435 registration pairs are generated by pairing each image with every other image. Similarly, the OASIS test set (60 images) yields 1,775 registration pairs.

All images are subjected to standard skull stripping and contrast correction preprocessing, with no prior registration (e.g., affine or rigid transformations) applied. Image segmentation is performed using FreeSurfer software (Fischl, 2012), resulting in 36 regions of interest (ROIs). Performance evaluations are based on ROI overlap in test images. For comparison, a separate training and testing dataset is created using affine registration with FSL's `flirt` command.

### 4.2. Training Details

Our models are trained using the Adam optimizer on a single NVIDIA A100 GPU. To determine the optimal hyperparameters, we employ a grid search strategy with a step size of 10, exploring various combinations of coefficients and learning rates. Based on this search, we set the initial learning rate to 1e-4 and reduced it by a factor of 0.5 every 50k iterations after the first 60k iterations for each level. The network is trained for 4, 4, 6, and 6 epochs in levels 1, 2, 3, and 4, respectively, ensuring robust convergence across all levels.

### 4.3. Evaluation metrics

To evaluate registration accuracy, we employ the Dice Similarity Coefficient (DSC) and Jacobian Determinant(JD). For the Dice computation, we calculate label-dependent Dice scores at the brain region level. The final Dice score is obtained by averaging the Dice scores across all labels. The Jacobian determinant of the deformation field is used to evaluate the local properties of the deformation. For a deformation field $\phi$, the Jacobian determinant $J_\phi$ at each spatial location measures the local volume change induced by the transformation. We also evaluate the computational efficiency of the registration method by measuring the processing time required for a single pair of images.

## 5. Results

### 5.1. Comparisons with the state-of-the-art methods

We compare our method with three widely-used registration approaches:

**SyN** (Avants et al., 2009) – A widely used registration method from ANTs, using cross-correlation and a multi-resolution optimization strategy with an initial affine transformation.

**ICNet** (Zhang, 2018) – An inverse-consistent deep network for unsupervised deformable registration, trained using the authors' optimal hyperparameters.

**LapIRN** (Mok and Chung, 2020) – A multi-level diffeomorphic registration algorithm using a Laplacian pyramid architecture and three CNNs, trained using the authors' optimal hyperparameters.

Table 1: Quantitative evaluation of different registration methods on IXI and OASIS datasets. Higher DSC values indicate better performance, while lower proportions of $|JD| \leq 0$ and registration times are preferred.

| IXI | SyN | ICNet | ICNet w.affine | LapIRN | LapIRN w.affine | Ours |
|---|---|---|---|---|---|---|
| DSC | 0.753 | 0.289 | 0.714 | 0.478 | **0.803** | 0.797 |
| | (±0.103) | (±0.211) | (±0.081) | (±0.272) | **(±0.069)** | (±0.137) |
| $|JD| \leq 0$ | **0.000** | 0.499 | 0.489 | 0.503 | 0.488 | 0.483 |
| | **(±0.000)** | (±0.002) | (±0.014) | (±0.041) | (±0.028) | (±0.027) |
| Time | 1 h | **0.252s** | 7.893 s | 1.100 s | 8.743 s | 0.272 s |
| | | **(±0.021)** | (±0.021 ) | (±0.006 ) | (±0.006 ) | (±0.042 ) |

| OASIS | SyN | ICNet | ICNet w.affine | LapIRN | LapIRN w.affine | Ours |
|---|---|---|---|---|---|---|
| DSC | 0.753 | 0.276 | 0.708 | 0.476 | 0.767 | **0.794** |
| | (±0.103) | (±0.184) | (±0.096) | (±0.182) | (±0.138) | **(±0.069)** |
| $|JD| \leq 0$ | **0.000** | 0.498 | 0.487 | 0.493 | 0.481 | 0.488 |
| | **(±0.000)** | (±0.002) | (±0.017) | (±0.042) | (±0.038) | (±0.042) |
| Time | 1 h | **0.232s** | 7.432 s | 1.670 s | 8.873 s | 0.266 s |
| | | **(±0.012)** | (±0.012 ) | (±0.051 ) | (±0.051 ) | (±0.054 ) |

Table 1 summarizes the quantitative results of our method and three comparison methods across all ROIs on two datasets. For a fair comparison, LapIRN and ICNet are trained and tested on both unregistered and pre-registered datasets. Our method achieves the highest Dice score on the unregistered dataset. While LapIRN achieved slightly better performance on the pre-registered IXI dataset, it incurs a significantly higher computational cost with only marginal improvement in accuracy. We conducted a statistical significance test and the relevant results are in Appendix B. We also test the dataset on two other baseline methods, Synthmorph(Hoffmann et al., 2021) and Gradicon(Tian et al., 2023), which can be found in Appendix C.

Figure 2 visually validates our method's registration precision, showing remarkable alignment with the fixed image without pre-registration. The intensity difference maps further substantiate the minimal residual misalignment.

Figure 3(a) showcases our network's robust reciprocal registration capabilities, particularly in handling subjects with substantial anatomical variations. The resulting deformation fields notably exhibit symmetric characteristics.

## 5.2. Ablation Study:

Figure 3(b) conducted ablation experiments by progressively removing key components of the model (gradient inverse consistency constraints, multilevel structure, symmetric structure, and attention-gate mechanism), which demonstrated the critical significance of each component in model performance: Employing gradient or symmetric consistency constraints

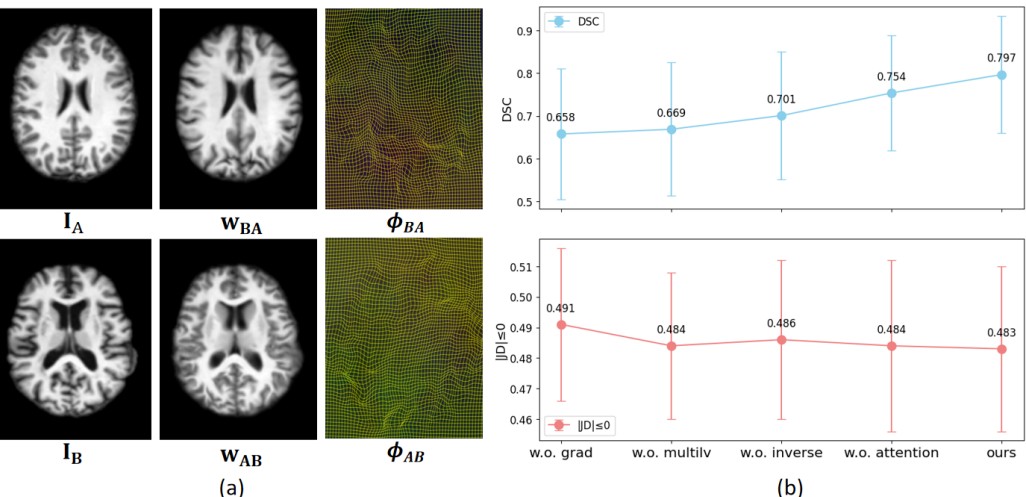

Figure 3: (a) The deformed images resulting from reciprocal registration of the image pair, and the resulting symmetric deformation fields. (b) Ablation study results show DSC and JD values obtained after removing each component of the network individually.

in isolation led to diminished registration accuracy; the absence of a multi-level network architecture substantially compromised the model's capacity to capture complex deformations; and integrating attention mechanisms at the highest resolution level enabled more discriminative feature representation across varying deformation scales, thereby preserving intricate feature information and ultimately enhancing registration precision.

## 6. Conclusion

We introduce a novel Symmetric Multi-level Gradient-Inverse Consistency Network (SM-GICNet) for robust large deformation image registration, specifically addressing the challenges posed by high inter-subject variability in medical images. Unlike many existing deep learning-based methods, SM-GIC Net directly handles large deformations without requiring pre-registration steps. This is achieved through a synergistic combination of three key innovations: 1) a symmetric multi-level architecture incorporating an attention gate mechanism for efficient multi-scale deformation capture; 2) a symmetric deformation field consistency strategy to ensure bidirectional symmetry and stability; and 3) a gradient inverse consistency constraint to reduce reliance on purely data- driven optimization and complexity.

## Acknowledgments

This work is supported by the National Natural Science Foundation of China [No. 32271146], the Startup Funds for Top-notch Talents at Beijing Normal University, China, and the fellowship of China National Postdoctoral Program for Innovative Talents. [No. BX20240039].

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

# Appendix A. Attention gate network architecture

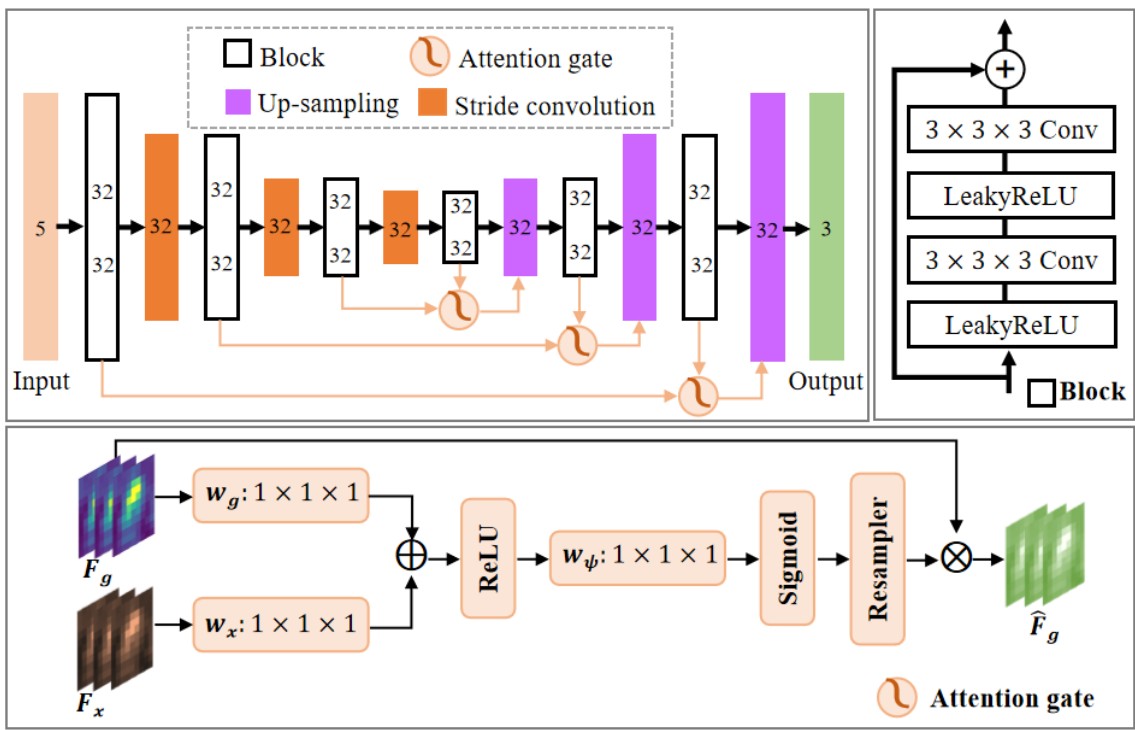

Figure 4: Attention gate network architecture.

The level 4 architecture retains an input layer, three residual blocks, and three convolutional layers with stride 2 in the encoder, along with three residual blocks, three upsampling layers, and an output layer in the decoder. Each residual block contains two consecutive convolutional layers. Three skip connections between the encoder and decoder incorporate attention gates. The attention gate first applies a 1x1x1 convolution to the input feature $F_g$ from the encoder and, similarly, to the downsampled feature from the corresponding encoder branch. These two outputs are then summed, followed by ReLU activation and another 1x1x1 convolution to reduce the channel dimension to 1. A sigmoid activation is applied to the result, resampled to match the original feature size, creating a 1D weight matrix. Finally, this weight matrix is multiplied with the input feature, producing a new feature map. This process enhances focus on registration-relevant local features, resulting in more accurate deformation fields and improved network convergence speed.

To validate its effectiveness, we experimented with networks of different channel sizes (e.g., 24, 32, 64) and found that 32 channels strike a balance between computational efficiency and registration accuracy. We also tested networks with varying depths and observed that simply increasing the number of layers does not improve accuracy.

## Appendix B. Statistical significance test

Table 2: Statistical significance test on IXI and OASIS

| Methods | IXI–Cohen's d | OASIS–Cohen's d |
|---|---|---|
| ICNet w.affine | 0.613 | 0.581 |
| LapIRN w.affine | 0.278 | 0.247 |
| ICNet | 3.113 | 2.691 |
| LapIRN | 1.578 | 1.387 |
| Ants | 0.253 | 0.227 |

We conducted a statistical significance test to evaluate the performance of our method compared to existing approaches, using the Dice coefficient as the primary metric. The results demonstrated that our method achieved a significant improvement over the comparison methods, as evidenced by Cohen's d (a parametric measure of effect size). Specifically, our method outperformed the baseline approaches without pre-registration, showing a substantial effect size that underscores its superior performance.

When compared to the LapIRN with pre-registration and SyN methods, our method achieved slightly higher significance in terms of Dice coefficient, while requiring significantly less computational time. This indicates that our approach not only delivers competitive accuracy but also offers a more efficient solution, making it highly suitable for practical applications where time constraints are critical.

In summary, the significance test highlights the effectiveness of our method in improving registration accuracy and its advantage in computational efficiency, positioning it as a robust alternative to existing state-of-the-art techniques.

## Appendix C. The other two baseline methods result

We have added two baseline tests, which are more recent. However, we can not train perfect models in a brief rebuttal, so we directly test using publicly available weights, with unsatisfactory results and numerous failure cases. It must be noted that these are excellent registration methods, but their research goals differ from ours, thus performing poorly in large deformation scenarios (especially without linear pre-registration).

**SynthMorph**(Hoffmann et al., 2021): Focuses on zero-shot learning, not specifically designed for large deformation registration.

**GradICON**(Tian et al., 2023): Emphasizes deformation field smoothness but has slow convergence in large deformation scenarios.

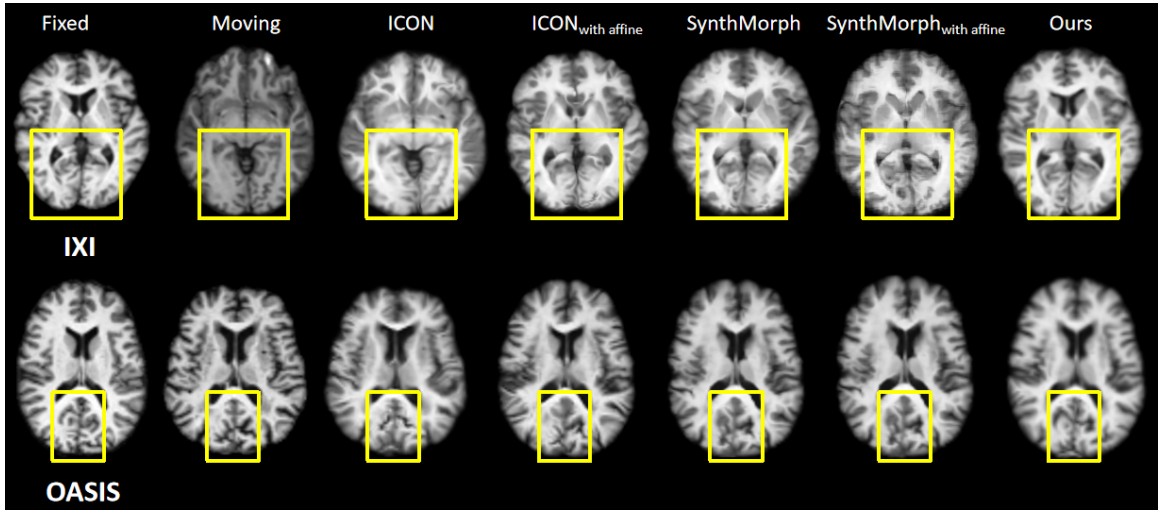

Figure 5: Qualitative registration results by two different methods and SM-GICNet on two datasets.

Table 3: Quantitative evaluation on IXI and OASIS datasets.

| IXI | ICON | ICON w.affine | SynthMorph | SynthMorph w.affine | Ours |
|---|---|---|---|---|---|
| DSC | 0.370 ($\pm$0.296) | 0.588 ($\pm$0.129) | 0.478 ($\pm$0.272) | 0.614 ($\pm$0.169) | **0.797 ($\pm$0.137)** |
| **OASIS** | **ICON** | **ICON w.affine** | **SynthMorph** | **SynthMorph w.affine** | **Ours** |
| DSC | 0.356 ($\pm$0.286) | 0.579 ($\pm$0.116) | 0.503 ($\pm$0.167) | 0.603 ($\pm$0.132) | **0.794 ($\pm$0.069)** |

