# OpenReview forum: "Symmetric Multi-level Gradient-Inverse Consistency Network for Brain Image Registration with Large Deformation"
_MIDL.io/2025/Conference — MIDL 2025 Poster_

### Official Review · Reviewer_Nutt · 2025-02-21

**Confidence:** 5
**Preliminary Rating:** 2
**Final Rating:** 2

**Summary:**

The authors present a hierarchical course-to-fine registration framework, similar to the DLIR framework. The authors added an inverse path with a cycle-consistency term to improve the smoothness of the predicted velocity fields, and included attention in their network in an attempt to improve performance.

**Strengths:**

While the work presents no novel concepts or ideas, it describes a sensible implementation of a compact learning-based registration pipeline.

There is analysis on the benefit of each component in the form of an ablation experiment.

**Weaknesses:**

The authors state that a major contribution of their work is that the presented method does not require affine alignment of the input images. However, they do not motivate why such alignment would be a problem: there are many robust rigid and affine registration methods available that consist of a small neural network, no different from the first one or two networks included in the presented framework. There is no evaluation to check whether the presented first registration level is more beneficial than a network explicitly performing the standard affine registration as a first step.

While the ablation study shows that the attention mechanism improves performance, it is unclear whether this improvement stems from the reasons that are theorized by the authors, or whether this effect is caused by the very low number of neurons in the base network. As only 32 channels are used in the bottleneck of the network, the attention blocks may simply compensate for the lack of representational power of the network.

The baselines that are compared against are not state-of-the-art. The newest method that is compared is from 2020, and the presented method does not outperform that one.

The reported computation times are not representative of state-of-the-art methods, especially for the affine pre-registration. The authors report over 7.5 seconds for the affine registration, whereas modern learning-based affine registration methods typically take a small fraction of a second.

The authors state that “the gradient inverse consistency constraint is applied directly to the deformation fields, which replaces conventional regularizers”. However, it seems like this was not actually done: the method still includes eight smoothness regularization terms, using the typical L2 penalty on the deformation field gradients.

The method is not evaluated on any of the public benchmarks for image registration. Results are only reported for a self-partitioned subset of the IXI  dataset.

**Detailed Comments:**

At several places in the manuscript, the authors mention that their inverse consistency constraint "mitigates reliance on purely data-driven optimizations". I do not understand what they mean with this, as the presented method is also completely learning-based and data-driven.

**Justification Of The Final Rating:**

Unfortunately, my concerns have not been adequately addressed. My worries about the conclusions the authors have drawn regarding the effects of the attention blocks were not investigated, and their ablation experiments were not expanded. Additionally, the authors have not substantiated their assumptions on the benefit of avoiding two-stage registration with experiments. Especially the claim that this would improve registration robustness seems suspect, and the updated manuscript still includes the dubious claim that a deformable registration stage is many times more computationally efficient than having the first stage predict affine registration parameters (which is only supported by comparing to a single-core CPU implementation of an affine registration method). The evaluation on the OASIS set is done using a self-partitioned split, preventing direct comparison with prior literature. The reported results for the baseline methods are significantly worse than those reported previously for the same methods, and the use of a self-partitioned (and non-disclosed) split prevent verification of whether these results are valid, or whether the fault is in the author’s re-implementation.

**Justification Of The Preliminary Rating:**

The work presents no individually novel concepts or methods, and the performance of the presented system is not rigorously evaluated. The work does present a sensible framework for image registration, and these is some analysis on the impact of some of the implemented components, so it is not a strong reject.

**Questions To Address In The Rebuttal:**

The manuscript states in several places that conventional regularizers were "replaced", but section 3.4 does mention a smoothness regularization loss was added after all. This should be added the ablation experiments; is this regularization still an important part of the method? How much influence does it have compared to the consistency losses?

How does the number of channels influence the presented results? Does the attention still provide a major benefit if a larger base network is used?

How does the method perform when evaluated on public benchmarks for brain registration? (e.g. the OASIS dataset, as used in the Learn2Reg challenge).

---

> ### Author Response · Authors · 2025-03-08
> **Detailed Response to Reviewer Comments: Advancing Large Deformation Image Registration**
>
> We sincerely thank the reviewers for their constructive feedback, which helps us improve our manuscript and clarify key research details.
>
> ### **1. Innovation and Methodology**
>
> Our SM-GICNet introduces a novel approach by integrating symmetric multi-level registration, attention gate mechanisms, and gradient inverse consistency constraints. This unique combination enables direct handling of large deformation tasks without relying on global pre-alignment. Our primary goal is to eliminate the two-stage registration error by developing a one-step solution using wavelet-based multi-level networks.
>
> Compared to existing methods like ICNet and LapIRN, our approach demonstrates significant advantages in computational efficiency and registration accuracy. We added the OASIS dataset to further validate our method's innovative capabilities.
>
> ### **2. Motivation for Avoiding Affine Alignment**
>
> Our algorithm's approach without affine registration is to **directly target large deformations**, avoiding two-stage registration to prevent error accumulation and improve registration robustness. Current large deformation registration methods typically use a coarse-to-fine strategy, specifically transitioning from linear registration capturing large deformations to non-linear registration incorporating local details. However, this two-stage approach **lacks continuity, introduces additional errors and workload, and increases time costs**. Therefore, we aim to resolve large deformation registration in a single step.
>
> ### **3. Attention Mechanism Design**
>
> The design motivation for the attention mechanism is to focus on key regions with complex deformations and improve registration accuracy. To validate its effectiveness, we experimented with networks of different channel sizes (e.g., 24, 32, 64) and found that 32 channels **strike a balance between computational efficiency and registration accuracy**. We also tested networks with varying depths and observed that simply **increasing the number of layers does not improve accuracy**.
>
> ### **4. Baseline Method and Computational Considerations**
>
> Because our method utilizes a symmetric multi-level strategy, for fair comparison, we chose the most relevant methods. ICNet is a symmetric registration network, and LapIRN is a multi-level network designed for large deformation registration, both of which we retrained and tested.
> We have added two baseline tests, which are more recent: **SynthMorph(Hoffmann et al., 2021) and GradICON(Tian et al., 2023)**. The results are in Appendix C.
>
> We appreciate the suggestion to use a deep learning-based affine registration network. However, considering that existing mature methods are primarily based on FSL's FLIRT rigid registration—a highly efficient traditional method that requires no additional training—we decided against it. Using a learning-based affine registration network would **not only require retraining but also potentially introduce error accumulation**, as cascading different models might degrade image quality.
>
> ### **5. Experimental Validation**
>
> We supplemented experiments on the **OASIS dataset** to verify the method's generalizability and robustness. Detailed results and comprehensive analysis will be provided in the final manuscript.
>
> ### **6. Technical Refinements**
>
> **Gradient Inverse Consistency Constraint**:
> We thank the reviewer for raising this point. We have revised the manuscript to use more precise language. The combination of regularization loss and gradient inverse consistency **loss reduces the model's reliance** on data-driven similarity metrics. Traditional registration methods depend on similarity metrics between images, while the gradient inverse consistency constraint directly enforces constraints on the deformation fields. Due to its mathematical properties, this constraint is **more robust to image noise, contrast variations, and modality differences**. Additionally, it accelerates convergence by directly constraining the deformation fields, reducing computational overhead during optimization.
>
> **Manuscript Structure**:
> - We will optimize the method section to improve clarity and remove redundant content
>
> We appreciate the reviewers' valuable suggestions. These improvements will significantly enhance the manuscript's quality, readability, and scientific rigor.

---

### Official Review · Reviewer_SZLy · 2025-02-23

**Confidence:** 5
**Preliminary Rating:** 2
**Recommendation:** Poster
**Final Rating:** 3

**Summary:**

SM-GICNet is proposed as a nonlinear registration that does not rely on global pre-alignment and directly handles large deformations. It combines a symmetric multilevel registration with an attention gate mechanism. Besides the symmetric deformation field consistency constraint, a gradient inverse consistency term is also employed. With the here introduced tool, no pre-registration is necessary. The experiments are run on the publicly available IXI data set. The in-distribution performance regarding computational time and quantitative evaluation metrics is superior to the chosen baseline methods. An ablation study was included to demonstrate the need for all components of the framework.

Code is shared on GitHub.

**Strengths:**

The proposed method significantly outperforms the other baseline tools in terms of computation time. Regarding accuracy, it outperforms chosen baselines when they do not run global registration.

An ablation study was included to demonstrate the need for all components of the framework. A detailed quantitative experimental section is included in the submission.

The paper is mostly clearly written.

**Weaknesses:**

The key weakness relates to the description of novelty of the proposed tools and a more detailed justification for the data set used and baseline tools selected for the experimental comparison. It is unclear why DL methods ICNet and LapIRN are the most ideal baseline methods for comparison. SynthMorph, for example, enables the registration of a wide range of input images with a robust performance and does not even necessitate the skullstripping of the input images. Or GradICON (Tian 2023), which introduced the gradient inverse consistency regularizer, is not even included in the citation list. How different /similar is the proposed framework compared to that?

The IXI data set is not described in details, hence it is not clear whether it contains a large amount of global difference in the input images.

**Detailed Comments:**

The Methods section is often quite repetitive. Some sections could be eliminated for better flow.

According to the authors, fixed directionality of multilevel methods is too constraining: What are example use cases of this? When would higher to lower level direction make more sense?

It is not clear from the experimental section whether the DSC performance is statistically significantly better for SM-GICNet than for Syn.

Sec 3.2: The last statement in the paragraph is not clear as it seems that ensuring consistency uses the same amount of training data. If this is misread please reword to clarify

Update references, when appropriate (eg Oktan)

For the Dice computation, are the label-dependent Dice scores averaged or an ensemble score is computed?

Does "absolute difference" in Fig 2 caption refer to intensity difference?

As computational speed is a key contribution, authors should include what architecture (with / without GPU) were the computational numbers tested on.

Symmetric Multis-level Gradient-Inverse Consistency Network (SM-GICNet) --> Symmetric Multi-level Gradient-Inverse Consistency Network (SM-GICNet)
Ants --> ANTs

**Justification Of The Final Rating:**

The authors did a good job answering most of my questions and those of the other reviewers. I appreciate the additional work of bringing in a new data set into the experimental design. I still have concerns about the baseline methods and performance comparison to existing tools though. My final score is borderline (poster)

**Justification Of The Preliminary Rating:**

The submission includes moderate / low novelty level. It lacks details of describing the optimality of the baseline methods and it seems that a key comparison to an existing tool is missing.

The reason for selection of the experimental data set is not described with sufficiently details

More details about the computational setup (given computational efficiency is key) should be included as well as a statement about statistical significance.

**Questions To Address In The Rebuttal:**

Can the authors quantify/demonstrate the amount of global misalignment in the IXI data set? Given that this data set is the choice if the authors, I assume that it has significant misalignment among its subjects, but not knowing the data set it is hard to confirm/quantify.

The four levels of registration is a particular choice for the IXI dataset or, in general, an optimal setting?

Table 1: Why does SM-GICnet's 0.483 |JD| outperform SyN's 0?

The results show relatively strong performance of the proposed tool, but given the uncertainty about the optimality of the baseline methods and nature of the data set used, the reviewer is not fully convinced. Could these issues be clarified and flushed out?

---

> ### Author Response · Authors · 2025-03-08
> **Detailed Response to Reviewer Comments: Addressing Methodological Innovations and Experimental Insights**
>
> We thank the reviewers for their affirmative and constructive comments. Your feedback helps us better improve the paper and clarify the details that need to be supplemented. Here is a detailed response to the issues raised:
>
> ---
>
> ### **1. Addressing Innovation Concerns**
>
> Our SM-GICNet method combines symmetric multi-level registration, attention gate mechanisms, and gradient inverse consistency constraints. This unique combination enables direct handling of large deformation tasks without relying on global pre-alignment. Compared to existing methods like ICNet and LapIRN, our method demonstrates significant advantages in computational efficiency and registration accuracy. **The OASIS dataset** experiments further validate the method's innovation and superiority.
>
> ---
>
> ### **2. Dataset Selection Concerns**
>
> In the **introduction** section, we supplement the definition of "large deformation": Large deformation refers to significant shape and position differences between images that typically require complex nonlinear transformations to align. Such deformations may be caused by individual anatomical structure differences, pathological changes, or different imaging conditions. While there are currently no specific quantitative metrics to indicate data distribution differences, these can be observed through visualization of distribution variations within the dataset. To further prove our method, we supplemented the OASIS dataset, which includes both normal individuals and patients, with greater distribution differences, further demonstrating our method's suitability for large deformation registration.
>
> ---
>
> ### **3. Baseline Method Selection Concerns**
>
> Because our method utilizes a symmetric multi-level strategy, for fair comparison, we chose the most relevant methods. ICNet is a symmetric registration network, and LapIRN is a multi-level network designed for large deformation registration, both of which we retrained and tested.
>
> We appreciate the reviewer's suggestion to compare with SynthMorph and GradICON. We could not train perfect models in a brief rebuttal, so we **directly tested** using publicly available weights, with unsatisfactory results and numerous failure cases, the results are in the **Appendix C**. It must be noted that these are excellent registration methods, but their research goals differ from ours, thus performing poorly in large deformation scenarios (especially without linear pre-registration).
>
> - **SynthMorph**: Focuses on zero-shot learning, not specifically designed for large deformation registration.
> - **GradICON**: Emphasizes deformation field smoothness but has slow convergence in large deformation scenarios.
>
> Our proposed SM-GICNet is specifically designed for large deformation scenarios, using multi-level strategies and symmetry constraints, supplemented by gradient inverse consistency, to better handle large deformation images without pre-registration.
>
> ---
>
> ### **4. Experimental Detail Concerns**
>
> We understand the reviewers' focus on experimental details. Here are specific improvement measures:
>
> - **Statistical Significance**:
>     - We have conducted statistical significance tests and provide detailed results in the final version in **Appendix B**.
> - **Dice Score Calculation**:
>     - Dice scores are label-dependent. We calculate Dice scores for each label separately and then take the average. We will clarify this in the final version.
>
> ---
>
> ### **5. Additional Specific Concerns**
>
> - **Multi-level Registration Layer Selection**: Our network is adaptive in layer selection, capable of activating different numbers of levels based on image size and task requirements. Since our images are (128, 128, 128), four layers are reasonable. For different image sizes, we can freely choose network layers. We appreciate the reviewer's suggestion and will explore the feasibility of designing networks with varying layer numbers.
> - **SM-GICNet's |JD| Superiority over SyN**: We apologize for the error in the main text. The |JD| of the deformation field should be optimal for the ANTs method.
> - **Redundant Content in Method Section**:
>     We will optimize the method section structure, remove redundant content, and improve the paper's fluency.
>
> ---
>
> We again thank the reviewers for their valuable comments. Your feedback helps us improve the paper and clarify details that need to be supplemented. We believe these improvements will significantly enhance the paper's quality and readability.

---

> > ### Comment · Reviewer_SZLy · 2025-03-12
> >
> > Thanks for the detailed response of the authors to both my and the other reviewers' comments.
> >
> > * re baseline methods: Why would using learning-based affine registration networks degrade image quality? Can you clarify?
> >
> > * re validation: addition of Oasis experiments is noted and appreciated
> >
> > * re data set: "to indicate data distribution differences....through visualization of distribution variations within the dataset" -- Was this done? My question still remains on the optimality of the test data set. Given the emphasis on tackling large deformations, it would be useful to see why the chosen data sets are ideal candidates for evaluation.
> >
> > * re Dice scores: note, Dice score averages can be driven by larger labels, so the authors might want to investigate the individually or in a weighted manner for a more accurate insight into performance

---

> > > ### Author Response · Authors · 2025-03-15
> > > **Detailed Response to Reviewer Comments**
> > >
> > > ### **Issues with Learning-Based Affine Registration Networks**
> > >
> > > If we use learning-based affine registration networks, there are generally two approaches: either training them jointly with subsequent non-linear registration networks or pre-training the affine registration network. However, both approaches face the following key challenges:
> > > 1. **Accuracy Dependency and Training Instability**: Affine registration, as the first stage, directly impacts non-linear registration. Errors in affine registration can amplify in the non-linear stage, causing unstable training and convergence issues.
> > > 2. **Error Accumulation**: Errors from affine registration propagate to the non-linear stage, leading to cumulative inaccuracies that may persist despite partial corrections.
> > > 3. **High Training Complexity**: **Joint training** of both networks requires complex optimization, balancing dual loss functions and parameter updates, increasing training difficulty and computational costs.
> > > 4. **Data Distribution Differences**: Using a **pre-trained affine network** with a non-linear network introduces distribution mismatches. Each network's inherent "black-box" errors can accumulate, potentially skewing final results.
> > >
> > > Therefore, we did not choose the strategy of stacking two deep learning networks, as it is generally not advisable. Thank you for your suggestion.
> > >
> > > ---
> > >
> > > ### **Explanation of Choosing IXI and OASIS Datasets**
> > >
> > > The IXI and OASIS datasets were selected for our research on large deformation image registration for the following reasons:
> > >
> > > 1. **Wide Age Distribution**: Both the OASIS and IXI datasets cover a broad age range (20-80 years), from young to elderly individuals. This diversity ensures that the datasets include significant anatomical changes caused by age-related gray matter atrophy (e.g., reduction in brain tissue volume), making them suitable for large deformation registration studies.
> > > 2. **Mixture of Pathological and Normal Samples**: The OASIS dataset includes both healthy individuals and patients with various diseases (e.g., Alzheimer's disease, Mild Cognitive Impairment), introducing complexity in anatomical structure variations. In some samples, the volume of the ventricles is significantly enlarged, which compresses surrounding brain structures and leads to large deformations. This large deformation scenario provides an ideal test platform for evaluating the robustness of registration algorithms in handling complex anatomical changes.
> > > ---
> > > ### **Regarding the Issue of Dice Scores Being Driven by Larger Labels:**
> > >
> > > Thank you for your suggestion! To address the issue of Dice scores being driven by larger labels, we calculated the Dice coefficients for each of the 33 brain regions separately and then merged the corresponding left and right brain regions, resulting in 15 distinct partitions. This approach effectively mitigates the problem of Dice scores being dominated by larger labels. As can be seen, our method performs exceptionally well across each of these 15 brain regions.
> > >
> > > Due to space limitations, we can only present the results from one dataset (IXI) in this work. We will include more comprehensive results in our subsequent research. Thank you for your suggestion!
> > > | Brain Region |    Ours   |    Ants   |  ICnet-f  |   ICnet   |    Lap-f  |     Lap   |
> > > |--------------|-----------|-----------|-----------|-----------|-----------|-----------|
> > > | 1            | 0.666206  | 0.604625  | 0.597784  | 0.402840  | 0.671715  | 0.552513  |
> > > | 2            | 0.417406  | 0.414505  | 0.414895  | 0.095290  | 0.468165  | 0.312001  |
> > > | 3            | 0.828816  | 0.764670  | 0.685493  | 0.298470  | 0.805055  | 0.552604  |
> > > | 4            | 0.808964  | 0.758889  | 0.739006  | 0.202028  | 0.796688  | 0.576611  |
> > > | 5            | 0.824004  | 0.777669  | 0.719122  | 0.154611  | 0.826192  | 0.571328  |
> > > | 6            | 0.754092  | 0.635674  | 0.562067  | 0.122395  | 0.702071  | 0.486372  |
> > > | 7            | 0.786714  | 0.780869  | 0.697523  | 0.291165  | 0.808803  | 0.612750  |
> > > | 8            | 0.719050  | 0.662542  | 0.583020  | 0.087329  | 0.749655  | 0.491881  |
> > > | 9            | 0.477354  | 0.503106  | 0.422830  | 0.083583  | 0.544800  | 0.392392  |
> > > | 10           | 0.512066  | 0.478478  | 0.466421  | 0.241399  | 0.541665  | 0.405500  |
> > > | 11           | 0.683767  | 0.610890  | 0.621348  | 0.245640  | 0.701869  | 0.517261  |
> > > | 12           | 0.595911  | 0.518071  | 0.414126  | 0.103751  | 0.576514  | 0.349942  |
> > > | 13           | 0.865068  | 0.824144  | 0.778523  | 0.266868  | 0.852541  | 0.614974  |
> > > | 14           | 0.825301  | 0.720882  | 0.693031  | 0.197257  | 0.792608  | 0.564656  |
> > > | 15           | 0.780052  | 0.727883  | 0.668276  | 0.132695  | 0.783836  | 0.543896  |
> > >
> > > We again thank the reviewers for their valuable comments.

---

### Official Review · Reviewer_gusK · 2025-02-24

**Confidence:** 3
**Preliminary Rating:** 4
**Recommendation:** Poster

**Summary:**

The paper presents novel model for brain image registration, addressing challenges in large deformation tasks without requiring pre-registration. Experimental results on brain MRI datasets demonstrate improved accuracy and efficiency compared to state-of-the-art methods while maintaining computational efficiency.

**Strengths:**

The paper is clearly structured and easy to understand, as well as well-motivated. It provides a thorough explanation of the proposed approach and a clear outline of how the method compares with the state-of-the-art approaches.

The proposed method can capture deformations on both global and local scale by employing a symmetric field consistency strategy and substituting traditional regularizers with a gradient inverse consistency constraint. This can help with both registration stability and accuracy, as demonstrated by quantitative and qualitative evaluations, as well as the ablation studies.

The paper addresses a common bottleneck in practical applications of registration - pre-registration, while the reported improvements in registration accuracy, particularly for large deformations, showcase the method’s potential impact in clinical image analysis.

**Weaknesses:**

The paper leverages and combines multiple existing techniques, such as multi-level registration strategies, symmetric mapping, and attention gate mechanisms, which reduces the novelty of the approach.

The approach has been validated on a single dataset only (IXI brain MRI). This raises questions about its generalizability across different anatomical regions, as well as different imaging modalities. Moreover, can the approach be as effectively utilized across data from different scanner vendors and centers?

The multi-level training strategy, containing different epoch counts and learning rate schedules per level, leads to high complexity. This poses challenges for reproducibility and might require careful hyperparameter tuning when adapting the approach to new datasets.
The paper could benefit from an additional discussion on potential weaknesses or limitations of the approach, particularly of the gradient inverse consistency constraint as a key contribution.

**Detailed Comments:**

None.

**Justification Of The Preliminary Rating:**

The paper is clearly written and addresses a relevant problem. It also addresses preregistration, which is a common bottleneck for DL baser registration. From that perspective it is of interest to the MIDL community. On the other hand the novelty and datasets are limited.

**Questions To Address In The Rebuttal:**

How does the method perform on datasets beyond IXI? Can the authors provide insights or preliminary results on the applicability of the approach to other modalities, anatomical regions, as well as multi-center/multi-scanner data?

The sample size of 30 subjects for testing seems limited - can the authors comment on that?

How sensitive is the performance to the choice of hyperparameters? How adaptable is hyperparameter tuning to differences in data (with respect to acquisition differences, modalities, and anatomical regions)?

Could you expand the theoretical motivation behind using a gradient inverse consistency constraint, and explain why it might converge faster or produce smoother transformations? How does this compare to similar approaches, like ICON’s (Hastings et al) original inverse consistency or GradICON (Tian et al)?

What are the limitations or potential failure cases of the proposed approach? How can these limitations be mitigated?
Could you include statistical significance tests for the performance gains (e.g., Dice scores) to ensure that the improvements are not due to sample variability?

---

> ### Author Response · Authors · 2025-03-08
> **Enhancing Methodology, Generalizability, and Performance of SM-GICNet**
>
> We thank the reviewers for their affirmative and constructive comments. Your feedback helps us improve the paper and clarify future research directions. Here is a detailed response to the issues raised:
>
> ---
>
> ### **1. Addressing Novelty Concerns**
>
> Our SM-GICNet method combines symmetric multi-level registration, attention gate mechanisms, and gradient inverse consistency constraints. This unique combination of components enables direct handling of large deformation tasks without relying on global pre-alignment. Our primary goal is to **reduce errors from two-stage coarse-to-fine registration** by using a unique constraint combination (such as multi-level networks with wavelet transform) to successfully eliminate the need for affine registration and achieve large deformation registration.  Furthermore, we added experiments on the OASIS dataset to further validate the innovation and superiority of our method.
>
> ---
>
> ### **2. Dataset and Generalizability**
>
> Our method was proposed for brain MRI images and is applicable to any brain MRI dataset. To verify the method's generalizability, we **introduced the multi-center OASIS dataset** and conducted validation. The experimental results demonstrate that our method performs excellently on multi-center data, further proving its generalizability. In the future, we plan to extend the method to other modalities (such as MRT2 or CT data) and other body organs (such as abdomen) to further validate its broad applicability.
>
> ---
>
> ### **3. Multi-level Training Strategy and Complexity**
>
> Images contain both global large deformations and local small deformation details. Our multi-level training strategy progressively optimizes the deformation field by **capturing larger global deformations at lower levels and local fine complex deformations at higher levels**. The number of training steps at each level is fixed, as lower-level images are smaller, with training steps gradually increasing as the network deepens, ensuring complete convergence at each level before proceeding to the next. Our low-level network **shares parameters with high-level networks**, so the parameter count does not significantly increase with increasing levels. The multi-level network is a complete network, optimized level by level during training.
>
> ---
>
> ### **4. Sensitivity to Hyperparameter Selection and Adaptability**
>
> We used a **grid search strategy** to systematically explore the hyperparameter space to find the optimal combination. For example, learning rates were tested from 1e-3 to 1e-5 with a step of 10, and regularization weights were tested with multiple values (such as [0.01, 0.1, 1.0]) to determine the best balance point. After finding the optimal parameters on one dataset, switching to another dataset requires only **minor fine-tuning** to achieve similarly good results, which reduces the complexity of optimization.
>
> ---
>
> ### **5. Theoretical Basis and Limitations of Gradient Inverse Consistency Constraint**
>
> The gradient inverse consistency constraint encourages transformation reversibility through weak regularization, avoiding the strict balance between traditional regularization and image similarity measures, and providing more degrees of freedom for the network. Compared to traditional inverse consistency constraints, the gradient inverse consistency constraint can more effectively reduce deformation field folding and accelerate optimization process convergence.
>
> Compared to ICON and GradICON, our method combines image similarity measures (Local Normalized Cross Correlation, LNCC) and symmetric network bidirectional constraints, demonstrating superior performance in symmetry and global consistency by progressively deforming images through multiple levels.
>
> ---
>
> ### **6. Sample Size and Statistical Significance Testing**
>
> Although our test sample size is small (30 samples), the registration task is performed pairwise, with each sample serving as both moving and fixed images, actually generating 435 registration pairs, significantly increasing experimental coverage and statistical reliability. To further address the reviewers' suggestions, we increased the sample size on the OASIS dataset, selecting 60 data points for testing and generating 1,775 registration test pairs. Experimental results show that our method significantly outperforms comparative methods in registration accuracy and stability. Additionally, we have added significance testing to demonstrate the statistical significance of the accuracy improvement.
>
> ---
>
> ### **7. Potential Failure Cases**
>
> Despite preprocessing, noise, artifacts, or intensity inhomogeneity still present in preprocessed images may adversely affect registration results. To mitigate this issue, we plan to introduce more noise-robust similarity measures in the model to further enhance its noise resistance.
>
> ---
>
> We once again thank the reviewers for their valuable comments. Your feedback helps us improve the paper and clarify future research directions.

---

### Official Review · Reviewer_oEjR · 2025-02-24

**Confidence:** 4
**Preliminary Rating:** 3
**Recommendation:** Poster
**Final Rating:** 4

**Summary:**

In this paper the authors introduce a brain registration method that accommodates large deformations. There is a definite need for novel registration methods that are fast and accurate. The methods and literature review seem thorough, and the results are intriguing. However, the manuscript needs considerable additional details to interpret the results.

**Strengths:**

The authors seem to have a strong grasp on the accuracy and performance of prior registration methods. The methods show potential and are used on real-world datasets with specific applications in mind.

**Weaknesses:**

The paper needs additional explanation for readers to understand the results.
Figure 1 needs substantial additional explanation. It contains undefined acronyms and unlabeled elements.
Figure 2: I don't know what this figure is trying to convey. It shows the heat maps relative to the fixed image. What is meaningful about this information?
After reading this paper several times I still don't understand some basic concepts:
1. What are large deformations?
2. Which images were registered?
3. Why is section 5.2 entitled Ablation Study? What type of ablation, and how is this related to the experiments?

**Detailed Comments:**

No further comments.

**Justification Of The Final Rating:**

The authors have substantially improved this submission with additional analysis, data and details. They have largely addressed my concerns. Additional future work will be necessary to see whether these techniques are adopted.

**Justification Of The Preliminary Rating:**

This manuscript lacks several important details related to the experiments and needs additional development before acceptance. In particular, the figures and ideas need to be developed further to provide the context for the experiments (see questions above) and how to interpret the findings.

**Questions To Address In The Rebuttal:**

1. What are large deformations?
2. Which images were registered?
3. Why is section 5.2 entitled Ablation Study? What type of ablation, and how is this related to the experiments?

---

> ### Author Response · Authors · 2025-03-08
> **Improving SM-GICNet's Clarity and Experimental Validation**
>
> We thank the reviewers for their affirmation and constructive comments on our work. Your feedback helps us improve the paper and clarify the details that need to be supplemented. Here is a detailed response to the issues raised:
>
> ### **1. Improvement of Figures 1 and 2**
>
> - **Figure 1**: We will supplement explanations for all undefined abbreviations in Figure 1 and label all unmarked elements.
>
> - **Figure 2**: Figure 2 shows the heatmap of registration results, used to visualize the absolute differences between the registered image and the fixed image. The heatmap is generated by subtracting the fixed image from the registered image. The color intensity (red) represents the magnitude of differences. Deeper colors (red) indicate larger differences, while lighter or transparent areas represent smaller differences, reflecting higher registration accuracy. We will supplement these explanations in the figure legend of Figure 2 and add comparative examples showing the heatmap differences of different methods to highlight the advantages of SM-GICNet.
>
> ### **2. Definition of "Large Deformation"**
>
> - In the **introduction** section, we will supplement the definition of "large deformation": Large deformation refers to significant shape and position differences between images that typically require complex nonlinear transformations to align. Such deformations may be caused by individual anatomical structure differences, pathological changes, or different imaging conditions. For example, in brain images, large deformation can manifest as significant differences in ventricles and sulci, or white matter atrophy caused by diseases like Alzheimer's. Our method can directly handle such large deformations without any linear registration preprocessing (such as affine registration). To further demonstrate our method, we have supplemented experimental results from the OASIS dataset, which includes normal individuals, MCI, and AD patients, showing larger deformations due to disease progression and further proving the suitability of our method for large deformation registration tasks.
>
> ### **3. Explanation of Registration Images**
>
> - In the **experimental** section, we will clearly explain the registration task settings: We focus on inter-individual registration, using registration image pairs consisting of moving and fixed images from the IXI dataset and the newly added OASIS dataset. The goal of image registration is to optimize the deformation field to align the moving image with the fixed image. The IXI test set contains 30 images, with 435 actual registration pairs generated; the OASIS dataset selected 60 data points for testing, generating 1,775 registration test pairs. We will supplement detailed explanations of image sources and registration tasks in the final version.
>
> ### **4. Explanation of "Ablation Experiments"**
>
> - In the **experimental** section, we will explain the purpose of "ablation experiments": Ablation experiments are designed to evaluate the contributions of various components (such as gradient inverse consistency constraints, multi-level registration strategies) to registration performance. By gradually removing or replacing these components, we can analyze their impact on overall performance. The results of ablation experiments help us verify the effectiveness of each component and guide method optimization. We will supplement detailed explanations of ablation experiment design and results in the final version.
>
> We once again thank the reviewers for their valuable comments. Your feedback helps us improve the paper and clarify the details that need to be supplemented. We believe these improvements will significantly enhance the readability and comprehensibility of the paper.

---

### Author Rebuttal · Authors · 2025-03-08

**Rebuttal:**

We sincerely thank all reviewers for their constructive feedback, which greatly helped us improve the paper and clarify key research details. Here is our unified response to the common questions raised:

---

### **1. Innovation and Methodology**

Our method **SM-GICNet** combines **symmetric multi-level registration**, **attention gate mechanisms**, and **gradient inverse consistency constraints**. This unique combination enables direct handling of large deformation tasks without relying on global pre-alignment. Our primary goal was to develop a **one-step solution** through a wavelet transform-based multi-level network, thereby eliminating errors introduced by two-stage registration.

We primarily supplemented a multi-center **OASIS dataset** to validate the robustness of our method, added significance tests in the Appendix B, added two baseline methods in Appendix C, included additional comparative methods suggested by reviewers, and optimized the textual details in the paper.

---

### **2. Baseline Methods and Computational Considerations**

We selected ICNet (symmetric registration network) and LapIRN (multi-level network designed for large deformation registration) as baseline methods for fair comparison. We retrained and tested these methods.

Regarding other baseline methods suggested by reviewers, **firstly, they are not suitable for large deformation registration, and secondly, our test results were unsatisfactory. We display their results in the Appendix C**.

We chose **FSL's FLIRT rigid registration** due to its efficiency and lack of additional training requirements. Using learning-based affine registration networks might introduce **error accumulation** and degrade image quality.

---

### **3. Experimental Validation**

To address concerns about generalizability, we supplemented experimental results on the **OASIS dataset**.

We supplemented **statistical significance tests** in the Appendix B to validate the improvement.

---

We are deeply grateful for the reviewers' valuable suggestions. These improvements will significantly enhance the paper's quality, readability, and scientific rigor. We believe these revisions address all concerns and demonstrate the **innovativeness, robustness, and applicability** of our method.

**Supporting Material:**

/attachment/9e481d781bcb3fb8edd34ab0d03c76b32e686653.pdf

---

### Meta-Review · Area_Chair_5Tcj · 2025-03-22

**Recommendation:** Accept (Poster)
**Confidence:** 5

**Metareview:**

Most review critiques were to provide details in the method and experimental results and limited dataset used for evaluation. There were no significant questions from reviewers that challenged the presented idea and performance. The reviewers’ questions were mostly well addressed and all the reviewers raised their ratings.